# Preparation and Performance Analysis of 3D Thermoformed Fluidic Polymer Temperature Sensors for Aquatic and Terrestrial Applications

**DOI:** 10.3390/s23208506

**Published:** 2023-10-17

**Authors:** Jahan Zeb Gul, Maryam Khan, Muhammad Muqeet Rehman, Zia Mohy Ud Din, Woo Young Kim

**Affiliations:** 1Department of Mechatronics and Biomedical Engineering, AIR University, Islamabad 44000, Pakistan; drzia@mail.au.edu.pk; 2Department of Electronic Engineering, Faculty of Applied Energy System, Jeju National University, Jeju 63241, Republic of Korea; maryamkhan93@stu.jejunu.ac.kr (M.K.); muqeet1988@jejunu.ac.kr (M.M.R.)

**Keywords:** graphite–glue polymer, temperature sensor, thermoforming, fluidic composite, underwater

## Abstract

Employing a combination of Polyethylene terephthalate (PET) thermoforming and 3D-printed cylindrical patterns, we carefully engineer a linear resistive temperature sensor. This intricate process involves initial PET thermoforming, yielding a hollow cylindrical chamber. This chamber is then precisely infused with a composite fluid of graphite and water glue. Ensuring electrical connectivity, both ends are affixed with metal wires and securely sealed using a hot gun. This cost-effective, versatile sensor adeptly gauges temperature shifts by assessing composite fluid resistance alterations. Its PET outer surface grants immunity to water and solubility concerns, enabling application in aquatic and aerial settings without extra encapsulation. Rigorous testing reveals the sensor’s linearity and stability within a 10 °C to 60 °C range, whether submerged or airborne. Beyond 65 °C, plastic deformation arises. To mitigate hysteresis, a 58 °C operational limit is recommended. Examining fluidic composite width and length effects, we ascertain a 12 Ω/°C sensitivity for these linear sensors, a hallmark of their precision. Impressive response and recovery times of 4 and 8 s, respectively, highlight their efficiency. These findings endorse thermoforming’s potential for fabricating advanced temperature sensors. This cost-effective approach’s adaptability underscores its viability for diverse applications.

## 1. Introduction

Polymer-based sensors hold considerable promise across industrial, clinical, and environmental applications due to their economical nature and straightforward fabrication processes [1,2,3,4,5,6,7,8,9,10,11,12,13]. They can be conveniently miniaturized and rendered disposable, all while being crafted from environmentally friendly and cost-effective substrates. The realm of sensor technology has been dynamically evolving with the integration of advanced fabrication techniques. Thermoforming, as a pivotal process, has garnered substantial interest for its potential to create versatile, customizable, and functional printed sensors tailored for applications such as temperature, strain, and pressure measurements [14,15,16,17,18]. This synthesis of thermoforming with sensor fabrication has paved the way for the development of sensors that not only exhibit enhanced sensitivity and accuracy but also offer intricate geometries that are challenging to achieve through conventional methods. Thermoforming holds a unique position in the landscape of sensor fabrication techniques. It involves the transformation of thermoplastic sheets over molds to create three-dimensional structures. This process not only allows for intricate sensor designs but also permits the integration of sensor elements into complex geometries. Unlike traditional two-dimensional (2D) fabrication techniques that are limited by their planar nature, thermoforming provides a bridge between the precision of three-dimensional printing and the adaptability of sensors to real-world applications [19]. The incorporation of printed electronics into thermoformed structures offers customizable, conformal, and stretchable 3D electronics. This integration enables sensors to seamlessly conform to irregular surfaces, enhancing their usability in various scenarios. Comparatively, thermoforming distinguishes itself from both two-dimensional and three-dimensional (3D) fabrication techniques. While 2D fabrication methods lack the ability to create complex geometries and 3D structures, thermoforming bridges this gap by enabling the creation of intricately designed sensors that can be seamlessly integrated into devices with curvatures, thus expanding their practical applicability. In contrast to 3D fabrication techniques like 3D printing [20], thermoforming stands out for its cost-effectiveness and scalability, making it conducive for large-scale sensor production. Thermoforming molds produced through 3D printing technology further contribute to the efficient creation of sensor prototypes and the rapid iteration of designs. These investigations highlight the multidisciplinary nature of thermoformed sensors, which merge materials science, electronics, and fabrication techniques to produce functional and adaptable sensor devices.

In this study, we employ a careful combination of Polyethylene terephthalate (PET) thermoforming and 3D-printed cylindrical patterns to engineer a linear resistive temperature sensor. This intricate process begins with PET thermoforming, which creates a hollow cylindrical chamber. This chamber is then precisely filled with a composite fluid consisting of graphite and water glue. To ensure electrical connectivity, both ends of the sensor are equipped with metal wires and securely sealed using a hot gun. This innovative and cost-effective sensor demonstrates remarkable versatility in accurately detecting temperature changes by analyzing alterations in the resistance of the composite fluid. Its PET outer surface provides inherent resistance to water and solubility issues, enabling its application in both aquatic and aerial environments without the need for additional encapsulation. Through rigorous testing, the sensor’s exceptional linearity and stability are evident within a temperature range of 10 °C to 60 °C, whether submerged or in airborne conditions. However, plastic deformation becomes apparent beyond temperatures of 65 °C. To address hysteresis concerns, a recommended operational limit of 58 °C is proposed. Further examination of the effects of fluidic composite width and length reveals a sensitivity of 12 Ω/°C for these linear sensors, showcasing their precision. Notably, the sensor’s response time of 4 s and recovery time of 8 s underscore its efficiency in real-time temperature monitoring applications. The results obtained from this comprehensive study highlight the promising potential of thermoforming as a viable method for fabricating advanced temperature sensors. The combination of 3D printing with thermoforming brings forth a multitude of advantages [21,22,23,24,25,26,27,28]. Firstly, it offers unparalleled customization, allowing the creation of intricately tailored designs for various applications, such as personalized medical implants. Secondly, this synergy permits the production of complex geometries that would otherwise be unattainable with traditional manufacturing techniques. Furthermore, it facilitates rapid prototyping, enabling quick design iterations and ultimately reducing time-to-market. Lastly, the cost-efficiency of this approach is notable, as 3D printing aids in the swift and cost-effective creation of molds, making it an ideal choice for both prototyping and mass production. The cost-effective nature and adaptability of this approach further emphasize its suitability for a wide range of applications across various industries.

## 2. Materials and Methods

### 2.1. Preparation of Fluidic Composite

Graphite, an allotrope of carbon, crystallizes into an ordered structure characterized by layers comprised of hexagonally arranged carbon atoms. The carbon atoms’ valence electrons display delocalization, granting them unrestricted mobility across the lattice matrix. This unique electron delocalization phenomenon is the cornerstone of graphite’s exceptional electrically conductive properties. Notably, graphite attains its peak electrical conductivity when measured parallel to the laminar layers, while its conductivity substantially diminishes along the perpendicular axis. The profound conductive attributes of graphite are a direct result of the uninhibited movement of electrons within its lattice structure, enabling the facile flow of electric charge. This variation in conductivity directions can be attributed to the differing binding forces acting upon the delocalized electrons within the graphite’s anisotropic arrangement. In an applied context, commercially available cheap graphite ink and water glue are purchased. Composition analysis using the Gravimetric method reveals a solid content of 11.46 wt.%. This method involves weighing a known amount (200 g in our case) of the ink, evaporating the solvent, and weighing the residue. The solid content is calculated as the ratio of the weight of the residue to the initial weight of the ink (11.46 wt.% in our case). This ink is blended with water-based glue at a ratio of 4:1, followed by magnetic stirring at 600 revolutions per minute for a duration of 20 min. The amalgamation of graphite ink and water glue initiates a process where the adhesive action of the water glue causes the dissolution of graphite particles, leading to their homogeneous dispersion within the mixture. This resultant fluidic composite, encompassing graphite and water glue, exhibits a noteworthy conductivity of 1.8 S/cm.

### 2.2. Sensor Fabrication

Thermoforming is a useful way to make things, especially when it comes to crafting 3D-printed sensors. The process involves heating up a plastic sheet (in our case, Polyethylene terephthalate (PET)) until it becomes soft and then molding it over a shape using either vacuum or pressure. When we use thermoforming for making 3D-printed sensors, it brings some clear advantages. First off, it lets us make sensors with complicated shapes very accurately and consistently. This means we can copy detailed designs precisely, making sure the final sensor matches what we planned. Plus, thermoforming gives us the freedom to choose from a variety of plastic materials that work well for sensors. This method suits different sensor types, like ones that measure temperature. On top of all that, thermoforming is quick and does not cost too much, which is great for making lots of sensors. Since 3D-printed sensors often need special shapes for specific uses, thermoforming combines the benefits of 3D printing and traditional making methods to ensure those intricate designs are just right. Thermoforming is highly cost-effective as compared to traditional fabrication techniques, including 3D printing. The step-by-step process of the fabrication of temperature sensor is illustrated in Figure 1.

Fabricating a temperature sensor involves a series of careful steps that seamlessly combine 3D printing and traditional manufacturing methods to achieve accurate and reliable results. The process begins by creating a cylindrical pattern through 3D printing, utilizing PLA material. Four samples are produced, each with varying dimensions (length cm/Dia mm), including 2/1, 2/2, 4/1, and 4/2, as mentioned in Table 1. The next phase entails the use of PET sheets, which undergo thermoforming by employing the 3D-printed patterns as molds. This process imparts a three-dimensional structure onto the PET sheet. Subsequently, the 3D-formed structure is extracted, leaving behind a hollow cylindrical pattern imprinted onto the PET sheet. Continuing the process, the PET sheets are joined together, leading to the creation of a hollow cylindrical chamber. This chamber serves as the fundamental structure that will house the sensor components. The fluidic composite is then inserted inside the hollow chamber using injection. This conductive mixture plays a crucial role in the sensor’s functionality, enabling accurate temperature measurements. Conductive wires are manually inserted inside the chamber openings to connect the fluidic composite with the external device. The chamber openings are then sealed using hot gun glue. This encapsulation ensures the sensor’s integrity and proper functioning. The composite inside the channel is fluidic and does not require any solidification or curing. Finally, the sensor’s resistive response is carefully gauged using a Digital Multimeter (DMM). This comprehensive approach allows for the accurate assessment of the sensor’s sensitivity to temperature fluctuations, ensuring its precision and reliability for various applications. The careful integration of these steps, from 3D printing and thermoforming to the strategic arrangement of components and comprehensive testing, culminates in the successful fabrication of a functional temperature sensor capable of providing accurate and dependable temperature readings.

### 2.3. Experimental Setup

After the successful fabrication of embedded sensing devices, a comprehensive investigation into the temperature-dependent resistance patterns was undertaken and carefully documented for further analysis. Precise electrical evaluations were conducted utilizing a custom-designed setup in-house, which allowed for the automated collection of sensor data. To establish a benchmark, a commercially available water-resistant temperature sensor (LM35: sensitivity 10 mV/°C, operating range up to 125 °C) was utilized. This reference temperature sensor was linked to an Arduino microcontroller, facilitating real-time data visualization and recording on a computer platform. The sensors produced were integrated into Arduino’s onboard analog-to-digital converter (ADC) in a potential divider configuration with a consistent fixed reference resistance. For investigations in dry environments, both the reference sensor and the fabricated sensor were placed on a heated plate, and the temperature was incrementally increased from 10 °C to 60 °C. Throughout this thermal progression, data from both the reference temperature sensor and the manufactured sensor’s resistance were continuously captured, logged, and visualized on a computer interface. To explore temperature reduction, the heated plate was substituted with ice packs, and corresponding data were collected. Readings at 0 °C were obtained by placing the sensors directly onto plastic ice packs. Multiple readings were taken for each device to ensure data precision and consistency. For data collection in aquatic environments, the sensors were physically moved between five distinct water containers, each maintained at a known temperature. Adequate time was allotted for the readings to stabilize, after which the resistance values were carefully documented. Furthermore, the sensors’ response and recovery times were recorded, necessitating rapid transitions between two liquids maintained at varying temperatures. This rigorous approach to data acquisition and analysis ensures a thorough understanding of the fabricated sensors’ performance under diverse environmental conditions.

## 3. Results

### 3.1. Material Characterizations

The conductive fluidic composite’s morphological and chemical attributes underwent comprehensive scrutiny employing field emission surface electron microscopy (FE-SEM) at 1 µm and 500 nm, FTIR, and Raman spectroscopy techniques. Field Emission Scanning Electron Microscopy (FE-SEM) is a highly advanced analytical technique that serves as a key instrument in examining the minute details of material surfaces at the nanoscale level. By employing a focused stream of electrons, FE-SEM scans the surface of a sample, generating exceptionally detailed images that unveil the intricacies of its topographical features. The significance of FE-SEM spans various scientific and industrial domains, as it provides invaluable insights into a material’s composition, distribution of particles, and particle sizes. The high-resolution images produced by FE-SEM enable researchers to unravel even the tiniest structural elements and characteristics, thus fostering a deeper comprehension of the material’s unique properties and behaviors. In order to perform SEM of fluidic composite material, a thin layer of fluidic composite is screen-printed on a PET substrate. For the fluidic composite of graphite and water glue, the lamellar structure is not observed in the SEM images, as shown in Figure 2a,b. A lamellar structure refers to a unique arrangement where materials consist of thin, flat layers stacked on top of each other. This layer-by-layer configuration often imparts distinct characteristics and behaviors due to the interactions between these sheets. Lamellar structures are observable in various natural and human-made materials, spanning from geological formations like sedimentary rocks to manufactured products such as specific polymers. After the ink/glue composites were introduced on the PET surface, a porous structure was found. This phenomenon can be attributed to the non-uniform volatilization of water in the ink and glue mixture, which leads to phase separation. The SEM images in Figure 2a,b show that the graphite embedded in water glue is uniformly dispersed.

Fourier Transform Infrared Spectroscopy (FTIR) is a key technique used to uncover the molecular intricacies and intermolecular bonds within the composite of graphite and water glue. By exposing the composite to infrared radiation and analyzing the resulting absorption and emission of energy, FTIR creates a unique spectrum that acts as a molecular fingerprint. This spectrum provides valuable insights into the chemical bonds and functional groups present in the composite, enhancing our understanding of its structural nuances. In Figure 2c, the fluidic composite of graphite composite and water glue shows distinct absorption peaks at characteristic wavelengths: around 3300, 2850−3000, 1750–1735, 1141, and 1150–1085 cm^−1^. These peaks correspond to specific molecular groups, such as O−H, C−H, C=O, C−O, and C−O−C, respectively. These insights into the molecular components give us a clearer picture of what constitutes the composite. Interestingly, these observations also indicate a strong interaction between water glue and graphite ink. The compatibility between the two materials is facilitated by hydrogen bonding interactions, particularly between the hydroxyl groups of the glue and the oxygen-containing groups of the graphite.

Exploring the chemical composition of the fluidic composite of graphite and water glue, Raman spectroscopy emerged as a pivotal technique for confirming the insights garnered and ensuring the consistency of the investigation. This technique utilizes laser-induced scattering of light to provide a deeper understanding of the molecular structure and interactions within the composite. As depicted in Figure 2d, the results of the Raman spectroscopy analysis reveal crucial graphene signature peaks located at 1576 cm^−1^ (G) and 1360 cm^−1^ (D). These peaks serve as distinctive markers that shed light on the unique arrangement of carbon atoms within the composite. In the context of the graphene–water glue composite, the G peak corresponds to the E2g mode, representing the in-plane vibrational motion of sp^2^-hybridized carbon atoms. The D peak, on the other hand, signifies the presence of defects or disorder in the lattice structure. This observation aligns with the Raman analysis of carbon-based materials, where the D peak’s intensity indicates the degree of disordered carbon atoms.

### 3.2. Electrical Characterizations

Precise electrical assessments were carefully executed through an in-house custom-designed setup, enabling automated data collection from the sensors. The results are shown in Figure 3. To establish a reference point, a commercially available water-resistant temperature sensor (LM35: sensitivity 10 mV/°C, operational range up to 125 °C) served as a benchmark. This reference sensor was linked to an Arduino microcontroller, offering real-time data visualization and recording on a computer platform. To explore the impact of length and cross-sectional diameter on sensing efficacy, four sensors of varying dimensions were fabricated, as mentioned in Table 1. Each sensing device underwent three rounds of readings to ascertain data reproducibility and statistical significance. The resistance of the sensors is measured underwater and in the air. For investigations in aquatic surroundings, the sensors were physically transferred between five distinct water receptacles, each maintained at a predefined temperature, as mentioned in Table 1. Adequate timeframes allowed the readings to stabilize before documenting the recorded resistance values. The response and recovery times of the sensors were also carefully noted, necessitating swift transitions between two liquids sustained at varying temperatures. This stringent approach to data acquisition and analysis underpins a comprehensive comprehension of the engineered sensors’ performance across a spectrum of environmental scenarios.

Among the array of fabricated sensors, it becomes evident that the sensor characterized by the most compact diameter, coupled with the most extensive pattern configuration, displays a distinctly elevated resistance. This sharp contrast is in stark comparison to the sensor with the most expansive diameter and the shortest pattern length, which exhibits the lowest resistance among its counterparts. This intriguing interplay between fundamental resistance values underscores a direct relationship with sensitivity, where a heightened resistance translates to augmented sensitivity and, conversely, reduced resistance corresponds to diminished sensitivity.

This linkage between resistance and sensitivity is quantifiable through the concept of absolute sensitivity in a linear sensor, which captures the change in resistance per unit alteration in temperature. Mathematically, this is succinctly represented by Equation (1), capturing the essence of this nuanced relationship. The engineered sensors, borne from careful fabrication, unveil a diverse spectrum of sensitivities. Notably, the pinnacle of sensitivity registers at an approximate value of 26 Ω/°C, a testament to the sensor’s capacity for rapid and precise response to temperature fluctuations. On the opposite end of this spectrum, the lower limits of sensitivity are observed at around 12 Ω/°C, reinforcing the variability and versatility of these sensors across various environmental conditions. These findings encapsulate the intricate connection between sensor geometry, resistance, and sensitivity, offering valuable insights for tailored sensor design and applications.
Sensitivity = (R_max_ − R_min_)/(T_max_ − T_min_)(1)

Upon careful examination of the plots depicted in Figure 3, a clear pattern emerges, revealing that the combined resistance of all devices exhibits a linear progression closely aligned with the gradual increase in temperature. This linear relationship holds steadfast regardless of the magnitude of temperature variation, underscoring the uniformity and coherence of the normalized curves across all sensors.

The comprehensive analysis further accentuates a discernible trend in the behavior of the curves, particularly as the length of the sensor increases while its diameter concurrently decreases. In this context, the curves progressively shift from being positioned beneath the ideal linear reference line to transgressing above it. These empirical observations collectively inform a nuanced understanding of how sensor dimensions intricately shape output characteristics. A comprehensive consolidation of these insights finds representation in Table 1, serving as an invaluable reference delineating the impact of varying sensor dimensions on their performance. The optimization of sensor performance hinges upon a careful calibration of pattern length and diameter. The operational mechanism underlying these sensors is intimately tied to the dynamic alterations that unfold within the conductive fluidic composite material. With the application of heat, the expansion of the conductive matrix within the fluidic composite engenders a reduction in conductivity due to the concomitant decrease in active contact points. Conversely, the cooling process prompts a reversion to the material’s more rigid state, subsequently restoring contact points to their inherent configuration. It is paramount to note that the operative range of these sensors is constrained by the inherent properties of the PET-based polymer matrix. Specifically, the maximum attainable operating temperature is delimited to 60 °C. Beyond this threshold, an irreversible deformation is initiated, leading to a deviation in resistance from its initial state. Given the proximate glass transition temperature of PLA, estimated at around 58 °C, a judicious operational range is recommended, encompassing temperatures up to, and ideally below, the 60 °C mark. This strategic temperature range offers the dual advantage of ensuring optimal outcomes while minimizing the effects of hysteresis—an indispensable consideration in temperature sensor applications.

The operational temperature range impressively lacks a detectable lower threshold, thereby endowing it with an extraordinary level of adaptability for a myriad of applications, spanning from terrestrial to subaquatic environmental and meteorological monitoring endeavors. This broad and encompassing operational spectrum distinctly highlights the sensors’ versatility, showcasing their adeptness in effectively capturing and quantifying temperature dynamics across a wide array of scenarios. For the careful assessment of stability and transient response characteristics, a specific sensor variant characterized by a length of 2 cm and a diameter of 1 mm was carefully chosen. The resistance values exhibited by this particular sensor fluctuate within a range spanning approximately 2 to 3 kΩ. A thorough evaluation of stability, depicted in Figure 4b, spanning a continuous operation period of 150 min, unequivocally underscores the unwavering nature of the sensors’ output, devoid of any discernible noise or uncertainty. In the investigation of hysteresis—an inherent characteristic impacting sensor performance—Figure 4a presents the recorded hysteresis in sensor readings. Employing Equation (2) to calculate the average percentage hysteresis, where ‘n’ signifies the number of samples, ‘yk’ denotes the resistance measurement at a given point, and ‘ymax’ and ‘ymin’, respectively, denote the maximum and minimum resistance values, the calculated value amounts to 4.93%. This careful analysis conclusively attests to the sensors’ possession of a negligible degree of hysteresis, well within the realm of acceptability. This marginal hysteresis impact inherently ensures precision and aligns seamlessly with the caliber of high-performance device developments witnessed in prior research.
(2)$$\mathrm{Average}\;\mathrm{Hysteresis}=[\;\sum_{k=0}^{n}\frac{y_{k+1}-y_{k}}{y_{max\;}-y_{min}}\;]$$

To verify the rapid response of the fabricated sensor, a transient response of more than five cycles is recorded, as shown in Figure 4c. The analysis of the transient response of two cycles, graphically represented in Figure 4d, divulges a noteworthy insight: the sensors demonstrate a rapid response time of 4 s for effecting a transition from 10% to 90% of their maximum value within the temperature span of 0 °C to 60 °C. The ensuing recovery phase, where the sensors revert to their initial values, registers a duration of 8 s. These temporal parameters align closely with those observed in advanced sensors highlighted within the scientific literature. This synchronization underlines the sensors’ capability to swiftly adapt and provide accurate outputs in response to temperature fluctuations. In the processing of data garnered from the fabricated sensors, an effective potential divider configuration interconnected with an Arduino-based microcontroller circuit was tactfully employed. This circuitry operates by converting digital data into corresponding voltages, subsequently translating them into sensor resistances, which in turn lead to temperature values. The pivotal transformation of the resistive response of a specific sensor into tangible temperature outputs was facilitated through Equation (3). Within this equation, ‘D’ represents the digital data output of the analog-to-digital converter (ADC), ‘Rref’ symbolizes the reference resistance, while ‘m’ and ‘c’ denote the slope and intercept, respectively. With a consistent reference resistance of 3 kΩ incorporated into the circuit design, the slope was carefully determined to be approximately 10, while the intercept was calculated to approximate 2200. This mathematical translation ensures accurate and reliable temperature determinations based on the sensors’ resistive behavior, ultimately facilitating their applicability in real-world scenarios.
Temperature = ((R_ref_ + c)D − (cD_max_))/(mD_max_ − mD) (3)

Illustrated in Figure 4e are the carefully delineated temperature-dependent output curves of the sensor, a culmination of applying the linear fit calibration curve equation within the microcontroller’s architecture. This pivotal mathematical conversion mechanism translates the acquired resistance values into their corresponding temperature measurements. In a sequential progression, the sensor’s response, intricately plotted alongside the data generated by a well-established commercially available reference sensor, stands as a testament to the device’s exceptional reliability and precision in quantifying real-world temperatures. The alignment between the sensor’s data and the reference sensor’s data confirms the sensor’s capability to deliver accurate temperature readings across a spectrum of practical scenarios, underscoring its robust and dependable performance.

## 4. Discussion

To provide a thorough contextual framework, a rigorous juxtaposition of the newly developed sensors with comparable and esteemed undertakings documented in the scientific literature is carefully furnished in Table 2. This extensive compilation encapsulates a succinct yet comprehensive overview of pivotal performance metrics, serving as a discerning tool to assess the sensors’ inherent quality. The table not only presents fundamental performance parameters but also delves into intricate facets such as device composition, the array of materials utilized, and the underlying operational principles that define their functionality. The strategic incorporation of data regarding analogous commercial sensors augments the breadth of perspective, offering a holistic vantage point to elucidate the nuanced prerequisites for the performance of application-specific sensors within this domain. An exhaustive examination of the tabulated data (Table 2) conspicuously highlights the current sensor’s remarkable emergence as a formidable contender, effectively challenging even the most illustrious sensors documented in prior instances of similar applications. The fabricated sensors’ resolution is 0.005, which is calculated using Equation (4).
Temperature Resolution = Smallest Detectable Change (SDC)/Scale Range (SR)(4)
where SDC is the reciprocal of sensitivity, and SR is 50. It is noteworthy that the sensor not only demonstrates superior sensitivity compared to a significant proportion of its predecessors but also exhibits a transient response profile that aligns itself with the most commendable benchmarks in the field and good resolution. Additionally, the sensor’s operational adaptability spanning both arid and aqueous environments provides a distinctive advantage that significantly outperforms the majority of its competing counterparts. Nevertheless, certain inherent limitations come to the forefront, encompassing a relatively narrower detection range and potentially nuanced precision when juxtaposed with commercially available devices.

## 5. Conclusions

The culmination of this study results in the innovation of an exclusive 3D-thermoformed temperature sensor carefully engineered for the purpose of environmental and meteorological monitoring. Notably, these sensors showcase remarkable compatibility with both arid and aqueous conditions. Demonstrating a commendable sensitivity level of 12 Ω/°C, these sensors are accompanied by swift transient response times of 4 and 8 s. Furthermore, they exhibit an impressively low hysteresis factor of approximately 4.9% while maintaining an inherently linear output curve and exceptional accuracy throughout the optimal operational temperature range, spanning from 10 °C to 60 °C. The materials employed in this study are easily accessible for seamless integration within conventional thermoforming setups. This eliminates the need for any excessive development efforts, be it in terms of the system or the materials themselves. The operational foundation of these sensors rests upon the temperature-triggered expansion of the fluidic composite, consisting of graphite and water glue. This expansion phenomenon is harnessed as a driving force in these sensors, leading to their response to environmental temperature variations. The stability and consistency exhibited by these sensors in their reactions to temperature fluctuations are highly pronounced, thus solidifying their efficacy. Furthermore, these sensors can be effortlessly incorporated into the intricate 3D computer-aided design (CAD) models of multifaceted devices such as 3D-printed robots or their various components. This unique capability enables a direct integration of these sensors into the structural framework of the ultimate device. This, in turn, empowers researchers with the potential to seamlessly embed these sensing modules within larger-scale structures devoid of any additional procedural complexities. This innovative approach streamlines the overall process and simultaneously opens avenues for the creation of pragmatic, fully functional robotic devices that are carefully tailored for potential applications in environmental and biomonitoring scenarios. The practical implications of this research are noteworthy, as it not only enhances the efficiency of sensing technology but also contributes to the realization of advanced robotic systems equipped for various challenging environments.

## Figures and Tables

**Figure 1 sensors-23-08506-f001:**
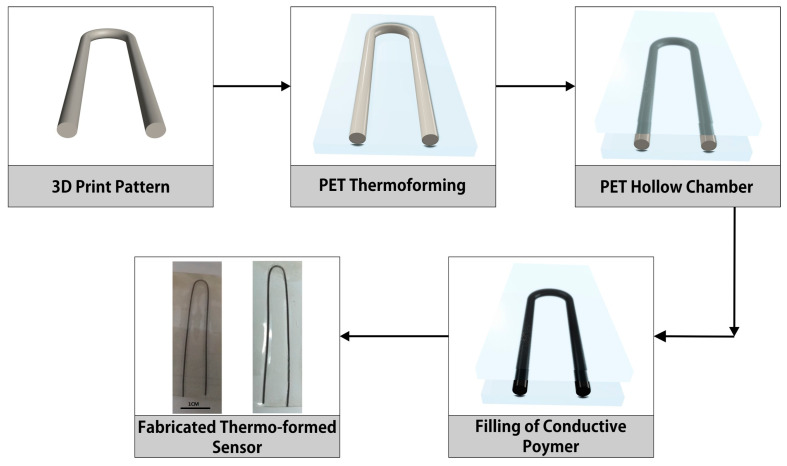
Step-by-step fabrication process of thermoformed 3D temperature sensor. In the first step, a cylindrical pattern is 3D printed. In the second step, PET substrate is thermoformed against 3D-printed pattern, resulting in a hollow pattern channel. In the third step, PET substrate layers are joined. In the fourth step, the hollow channel is filled with fluidic composite of graphite and water glue.

**Figure 2 sensors-23-08506-f002:**
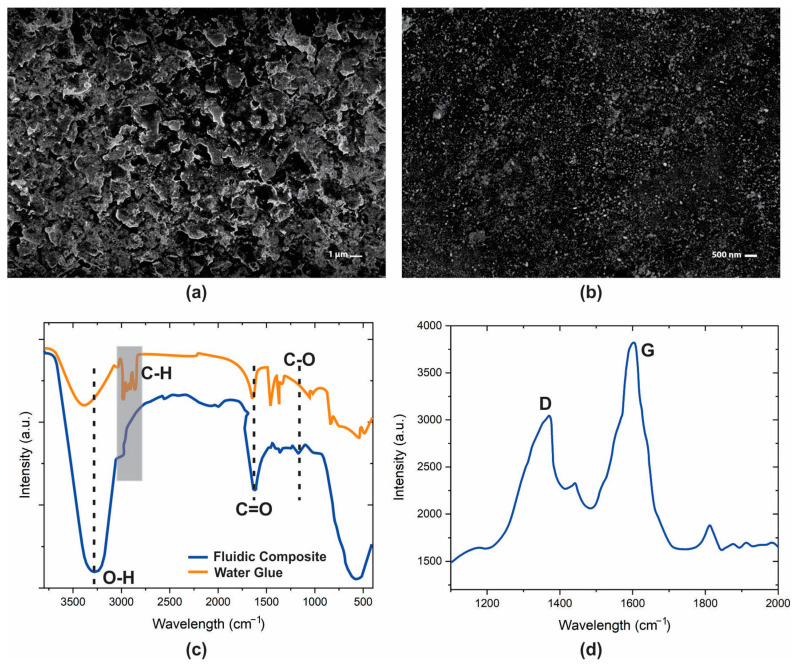
Material characterization results. (**a**) SEM image of the fluidic composite of graphite and water glue (4:1) with 1µm scale; (**b**) SEM image of fluidic composite of graphite and water glue (4:1) with 500 nm scale; (**c**) FTIR of fluidic composite and water glue; (**d**) Raman of fluidic composite.

**Figure 3 sensors-23-08506-f003:**
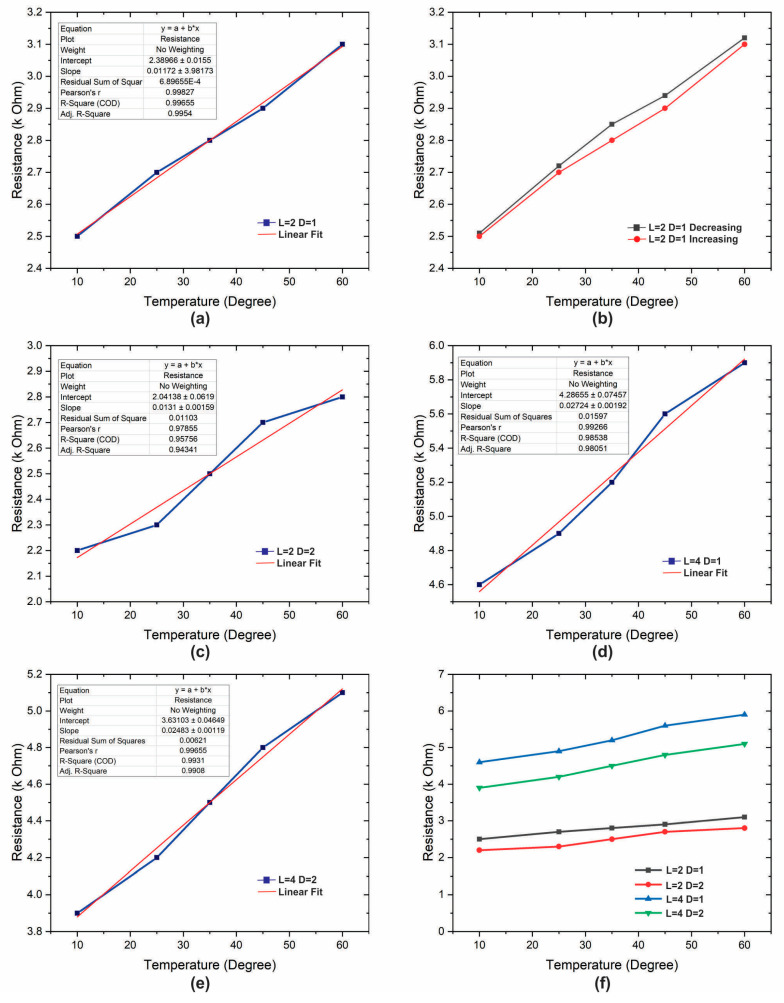
Resistive response of temperature sensor underwater and the effect of changing length and diameter of fluidic composite pattern. (**a**) Resistive response and linear fit of L2-D1; (**b**) hysteresis of L2-D1; (**c**) resistive response and linear fit of L2-D2; (**d**) resistive response and linear fit of L4-D1; (**e**) resistive response and linear fit of L4-D2; (**f**) cumulative resistive response of all four samples (L2-D1, L2-D2, L4-D1, L4-D2).

**Figure 4 sensors-23-08506-f004:**
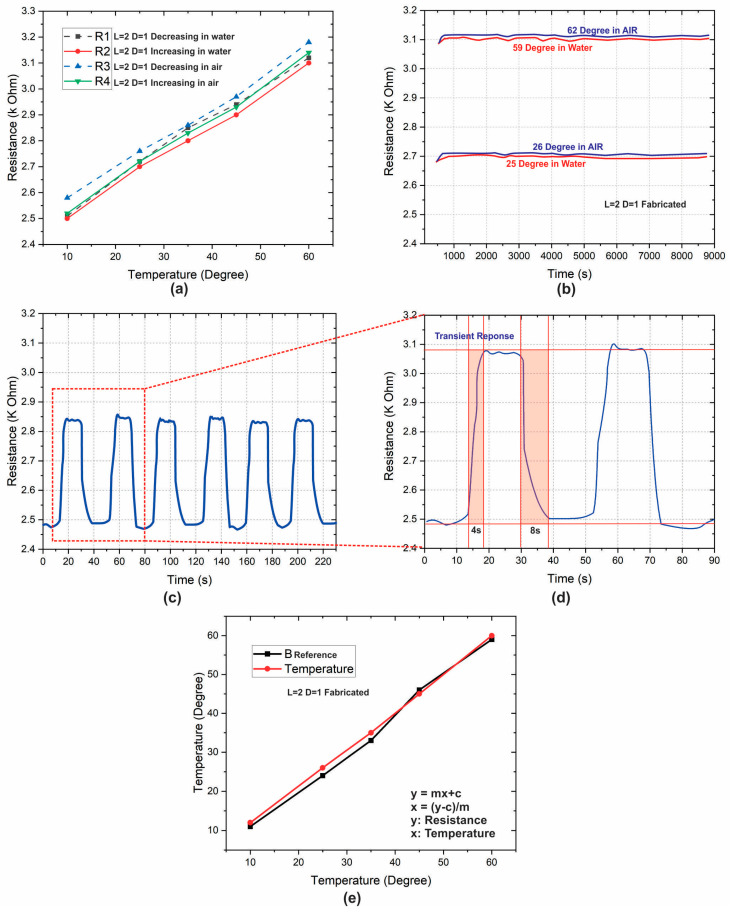
Performance evaluation of the fabricated thermoformed temperature sensor. (**a**) Hysteresis curves (increasing and decreasing) of temperature in air and underwater; (**b**) operational stability plot for continuous operation of 150 min; (**c**) transient response curve for multiple cycles (n > 5); (**d**) transient response curve of two cycles; (**e**) performance comparison with reference(commercial) sensor.

**Table 1 sensors-23-08506-t001:** Output with respect to mechanical parametric difference.

Length (cm)	Dia (mm)	Min. Res (kΩ)	Max. Res (kΩ)	Sensitivity (Ω/°C)
2	1	2.5	3.1	12
2	2	2.2	2.8	12
4	1	4.6	5.9	26
4	2	3.9	5.1	24

**Table 2 sensors-23-08506-t002:** Summary of literature review and performance comparison of temperature sensors.

Materials/Sensor Name	Fabrication Method	Max Detection Range (°C)	Sensitivity	Transient Response(s)	Operational Environment	Reference
TMP36	Lithography	120	10 mV/°C	20	DRY/WET	Commercial
LM35	Lithography	125	10 mV/°C	8 s	DRY/WET	Commercial
DS18B20	Lithography	100	12-bit resolution	NC	DRY/WET	Commercial
TMP37	Lithography	100	10 mV/°C	NC	DRY/WET	Commercial
GNR + PLA	3D Printing	70	12 Ω/°C	6 s	DRY/WET	[29]
Flake graphite/CNT/PDMS	Screen Printing	80	0.028%/°C	NC	DRY	[30]
Mn_2_O_3_/NiO/Co_3_O_4_/CuO/ZnOPVDF, PDMS, CYTOP	Screen Printing	140	91.76%	NC	DRY	[31]
Polyvinyl chloride/carbon black	Screen Printing	44	−0.148%/°C	NC	DRY	[32]
Graphite + Water Glue	3D Thermoforming	60	12 Ω/°C	4	DRY/WET	This work

## Data Availability

Partial data files are available at www.jahanzebgul.com (accessed on 15 September 2023). Complete data can be made available on reasonable request.

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
