# Peer review of "Preparation and Performance Analysis of 3D Thermoformed Fluidic Polymer Temperature Sensors for Aquatic and Terrestrial Applications"

_sensors, 2023, doi:10.3390/s23208506_

Round 1

Reviewer 1 Report

The authors present a straightforward and cost-effective 3D temperature sensor created through 3D printing and thermoforming techniques. The data and discussions within the manuscript are both compelling and thought-provoking, making it highly engaging for a wide range of readers. Minor comments as follows:

1. Including a scale bar in Figure 1's photo would greatly enhance its clarity.

2. Throughout the manuscript, there are multiple typos and missing spaces between numbers and SI units that need correction. For instance, there is a space between the number and the degree symbol for temperatures.

3. In Figure 2, consider using wavenumber on the X-axis for the spectroscopic data instead of wavelength.

4. It would be valuable to include FT-IR spectra for both water glue and graphite ink separately as part of a control experiment. This can help illustrate any hydrogen bonding interactions between the two materials, and validate the mixture of the two materials.

5. To further address the hysteresis effect, it is recommended to demonstrate the reliability of the output resistance over multiple cycles (n > 5) of increasing and decreasing the temperature, in addition to the data presented in Figure 4b.

The term "meticulous" is frequently reiterated in the text, and it might benefit from some synonym variation for improved readability.

Author Response

We sincerely appreciate your insightful and encouraging remarks regarding our manuscript. Your feedback has been both motivating and constructive. We tried our best to integrate the changes as per the minor comments raised by you.

Minor comments as follows:

  1. The scale bar in Figure 1 is now added.
  2. We made the necessary corrections and reviewed the manuscript in detail. However, detailed verification of typos and missing spaces will be done again when the manuscript is accepted and in the final production stage.

  3. Thank you for the suggestion but in our case, using wavelength provides a more intuitive representation that aligns with the common practice in our field according to reference papers. Therefore, we request to keep the wavelength on the x-axis for the spectroscopic data.

  4. Following your insightful recommendation, we have conducted the necessary experiments to obtain FT-IR spectra of water glue as well and now FT-IR of both water glue and fluidic composite is included in Fig.2(c).

  5. In response to your suggestion, we have conducted additional experiments involving more than five cycles of temperature variation. The data from these experiments have been analyzed, and the results are included in Fig.4(c) and 4(d).

  6. The term “meticulous” was mentioned with respect to careful experimentation throughout the manuscript. This term is now removed from the manuscript to avoid any confusion.

Reviewer 2 Report

This manuscript presents the use of 3D printing technology combined with thermoforming abrasives to prepare a resistive temperature sensor. In general, the manuscript is well-organized and written with innovative and creative ideas, therefore I think this manuscript is qualified to be published in a journal. In addition, there are some suggestions to can help improve the quality of the manuscript.

1. The content of this paper primarily revolves around a novel sensor fabrication process and the analysis of the resulting sensor's characteristics. Therefore, the title "Aquatic and Terrestrial 3D Thermoformed Fluidic Polymer Temperature Sensor" does not adequately convey the aspects of sensor fabrication and performance analysis. Hence, I suggest changing the title to "Preparation and Performance Analysis of 3D Thermoformed Fluidic Polymer Temperature Sensors for Aquatic and Terrestrial Applications" to better reflect the paper's theme and research content.

2. While the introduction of this paper effectively highlights the advantages of combining 3D printing technology with thermoformed molds for sensor fabrication, it lacks a comprehensive review of the literature in this field, such as the contributions and achievements of specific researchers. The conclusion of the introduction should also summarize the primary innovative aspects of this paper in comparison to the works of other scholars, avoiding the inclusion of conclusive statements typically found in the conclusion section.

3. Is the fluid of graphite and water glue used in the paper self-made or purchased as ready-made materials? This should be clarified within the text. While the paper provides a description of the properties of graphite, it lacks a detailed explanation of the properties of the water glue. Supplementary information regarding the attributes of the water glue should be included.

4. To my understanding, one of the key innovations in this work is the proposal of a novel sensor fabrication process, in which the composite fluid of graphite ink and water-based adhesive plays a significant role in the preparation. In this paper,  'Through meticulous experimentation' is used to describe this process. I believe it would be beneficial to provide a more detailed description of this process, such as comparing different experimental parameters or referencing prior work that demonstrates the optimal parameters for achieving electrical conductivity in the graphite and water-based adhesive composite fluid.

5. As is mentioned in line 137 of the paper, 'The subsequent step involves the introduction of a fluidic composite consisting of graphite ink into the hollow chamber.' After the introduction of this 'fluidic composite consisting of graphite ink' into the chamber, in what form does this material exist? Does it require curing or solidification? Additionally, how are the conductive wires inserted into the chamber in the presence of this material? Moreover, in line 140, it is stated, 'Conductive wires are meticulously inserted into the chamber, followed by the secure sealing of the PET sheet ends.' Encapsulation is a crucial step in sensor fabrication, however, the paper failed to provide specific details regarding the final steps of sensor encapsulation.

Author Response

We sincerely appreciate your positive assessment of our manuscript. We tried our best to make the necessary changes as per the comments raised by you. The responses to the comments are as follows:

  1. We greatly appreciate your thoughtful suggestion regarding the title of our manuscript. We agree with your assessment that the proposed title, "Preparation and Performance Analysis of 3D Thermoformed Fluidic Polymer Temperature Sensors for Aquatic and Terrestrial Applications," more accurately reflects the core focus of our paper, which revolves around sensor fabrication and performance analysis. In response to your recommendation, we have updated the manuscript title accordingly.
  2. We have expanded the literature review section in the introduction by incorporating more references and providing additional details on the contributions and achievements of specific researchers in the field of combining 3D printing technology with thermoformed molds for sensor fabrication. This addition will better contextualize our work and demonstrate its significance in relation to previous research.

  3. We appreciate your inquiry regarding the source of the fluid materials used in our study, specifically the graphite and water glue. To clarify this aspect within the text, we have made the following additions to the manuscript:

    • We have included a statement within the Materials and Methods section specifying that both the graphite ink and the water-based glue used in our experiments were commercially purchased. This information is now clearly mentioned to provide transparency regarding the source of these materials. The process of formation of fluidic composite is mentioned in detail as well.

    • In response to your request for more detailed information about the properties of the water glue, we have included the Fourier-transform infrared spectroscopy (FTIR) data for the water glue in Figure 2c of the manuscript. This FTIR data provides insights into the chemical composition and attributes of the water glue.
  4. The meaning of the term 'Through meticulous experimentation' is ‘careful experimentation’ only. In order to avoid confusion, the term meticulous is removed throughout the manuscript, and the process of fluidic composite formation is mentioned properly in the section “Preparation of Fluidic Composite Section”.

  5. The material exists in fluidic form in the chamber. It does not require any curing or solidification. The connecting conductive wires are manually inserted in the fluidic chambers and openings of the chambers are sealed using hot glue to keep the fluidic composite inside the channel. All this is now clearly included in the manuscript.

Reviewer 3 Report

The proposed sensor shows relatively good results regarding temperature sensitivity, transient response time, and ability to operate in a water-rich environment. However, suppose the proposed work is compared to commercially available thermocouples. In that case, its maximum temperature range and response time are not as good (0.25 mm thermocouple response time is as low as 0.015 s). And their operating temperature can easily go over 260 degrees Celcius. And the fabrication process requires some tedious assembly processes, which can increase the cost of sensors compared to the counterparts fabricated by lithography.

Miscellaneous items

1. The manuscript does not show how the composition analysis was performed and verified. Please explain the method used for the analysis and how the solid content of 11.46 wt. % was chosen.

2. The number of data points for the temperature response of the sensors needs to be increased to capture the characteristics of the sensors accurately.

3. If the temperature resolution is compared among the different sensors, it will be easy to compare their performances.

The quality of English writing looks okay to me.

Author Response

We sincerely appreciate your insightful feedback regarding the comparison of our proposed sensor with commercially available thermocouples. Our novelty is the use of the thermoforming technique to fabricate temperature sensor and our fabricated sensor also showed a respectable response in both water and air environments (Table-2). Thermoforming is extremely cheap and widely used in industry. In this study, we explored the possibility of using cheap material composite (Graphite+Water Glue) on PET substrate to fabricate a temp sensor. The importance and previous relevant work of thermoforming is mentioned in the introduction part in detail with references. Response to your suggestions is as follows:

  1. Graphite Ink is purchased locally and solid content in the ink (11.46%) is determined by the Gravimetric Method (weighing a known amount of the ink, evaporating the solvent, and weighing the residue). It is now clearly mentioned in the section “Preparation of Fluidic Composite”.

  2. We have conducted additional experiments involving more than five cycles of temperature variation. The data from these experiments have been analyzed, and the results are included in Fig.4(c) and 4(d). This proves the sensor’s performance and reliability. The number of data points for the temperature response of the sensor is selected based on previously published literature.

  3. Although detailed performance analysis with commercial and literature was included in Table-2 which shows the sensitivity data as well, keeping in view of your suggestion, the resolution of the fabricated sensor is computed, and the equation (Eq-4) used to compute the resolution is also mentioned in the manuscript for future reference.

Round 2

Reviewer 3 Report

The authors addressed all the comments adequately.